# Graph Attention Network generates Super-resolution Spatial Transcriptomic data

## Abstract

Spatial transcriptomic technologies allow for uncovering the spatial origin of RNA molecules within a tissue slide. Still, some challenges remain unsolved when acquiring informative signal. An existing trade-off hinders the choice of which one to use: sequencing-based technologies provide high-throughput profiles, while imaging-based outperform regarding spatial resolution. On the sequencing-based side, the minimal spatial unit, called spot, comprises more than one cell, yielding slightly blurred expression profiles. To avoid inaccurate analysis and misinterpretation of spatial data, we believe that cells inside a single spot should be isolated and allocated into subspots. We propose a computational method based on graphs and attention learning, named Square, that leverages message passing for information sharing between neighbor spots. Even though this rearrangement of cells can be solely spatially approximated, a resolution enhancement is achieved. We show that the proposed approach is capable of deciphering the composition of ST spots, whilst imputing sparse profiles and amplifying the signal in them. Newly generated subspots have been empirically and biologically validated. The gap between both spatial transcriptomic modalities is then closed, generating high-throughput cellular-scale outputs.

## 1 Introduction

The advent of spatial transcriptomics has enabled the precise mapping of gene expression profiles within tissue architecture (Moses & Pachter, 2022), providing key insights into cellular composition and thus, significantly advancing our understanding of the molecular signatures that drive disease progression and therapeutic responses (Piwecka et al., 2023; Arora et al., 2023). While imaging-based spatial transcriptomic technologies, which leverage the *in situ* hybridization of transcripts, can yield subcellular transcriptional resolution, they are constrained to a few thousand genes through custom gene panels (Rao et al., 2021). Sequencing-based technologies (such as 10X Genomics Visium, or Spatial Transcriptomics (ST)), on the other hand, offer an unbiased view of the transcriptional architecture of the tissue at a cost of cellular resolution, as their minimal spatial unit can contain several cells (e.g., 3-10 cells in the $55\mu m$-diameter spots in Visium). To address this critical issue, several computational models have been lately developed for deciphering the spot composition of different cell types, termed deconvolution models (Ma & Zhou, 2022; Andersson et al., 2020; Zhou et al., 2023; Lopez et al., 2021). However, recent benchmarking contributions depict a lack of consistency among state-of-the-art deconvolution methods and unexpected differences in their performance on different datasets (Yan & Sun, 2023; de la Fuente et al., 2023).

The very recent release of sequenced-based Visium HD (Oliveira et al., 2025) has challenged the current trade-off between (sub)cellular resolution and the unbiased transcriptome profiling by moving from Visium's $55\mu m$-diameter spots to Visium HD's $2 \times 2\mu m$ bins. However, the cost of using the proposed pipeline has increased compared to the standard Visium one (67% more expensive), with the reagent being the main culprit (74%) of such an increase (Smith et al., 2024). Additionally, the number of unique transcripts (i.e., Unique Molecular Identifiers (UMIs)) captured per bin is significantly lower, and so is the transcriptional information captured within. Hence, the sparsity levels of transcriptional signals, which stand as one of the main hurdles in the analysis of single-cell and spatial omics data, have increased even further (Kamel et al., 2025).

In order to overcome the currently scarce signal in the sequencing-based technologies, we believe that deconvolution can be conceptually extended to a spatial reallocation of transcripts from an initial spot to smaller subspots within it. This would translate into a spatial resolution enhancement, using only the available information from a spatial transcriptomics dataset. To this end, we propose Square, a self-supervised model based on a graph attention network that untangles spots' signals and generates super-resolution spatial transcriptomic data. We show that the enhanced subspots generated by Square: i) contain very low entropy cell-type proportions, ii) better capture the spatial location of cells within the original spot; and iii) uncover key biological insights that were hidden in the original data. All in all, we envision that the proposed model will guarantee a minimal spatial resolution on which different technologies can be systematically compared, hence reducing the effects of a progressive obsolescence of available data.

## 2 Towards spatial super-resolution in spatial transcriptomics

**Problem statement.** A sequencing-based spatial transcriptomic experiment generates a gene expression matrix $\mathbf{X} \in \mathbb{R}^{G \times S}$, containing the expression of $G$ genes across $S$ spots, and a coordinate matrix $\mathbf{Z} \in \mathbb{R}^{2 \times S}$, containing the coordinates $(x, y)$ of each spot. Our goal is to generate a *high resolution (hr)* sample by generating $k$ new subspots for each spot, producing a new expression matrix $\mathbf{X}_{hr} \in \mathbb{R}^{G \times kS}$ and corresponding coordinate matrix $\mathbf{Z}_{hr} \in \mathbb{R}^{2 \times kS}$, yielding an enhanced spatial map. In what follows, $k$ is set to 9, so that an isotropic expansion of both $x$ and $y$ axes is guaranteed.

**Self-contained spatial inference.** To facilitate the application of the developed model to complex tissues, our goal is to use only the provided spatial transcriptomics sample to generate these subspots, i.e., not requiring additional information such as paired single-cell RNA sequencing (scRNA-Seq) data or the number of configurations (e.g., cell-types) in the dataset. In this context, datasets conventionally include a standard microscopic image, most commonly hematoxylin and eosin (H&E)-stained. The introduction of subcellular resolution with VisiumHD has further elevated the relevance of these images in downstream analyses, as segmented cells constitute the primary processing units for this technology. This trend is reflected in the substantial increase in image resolution, with file sizes reaching up to 8 GB. Although the field has begun to explore the use of such images for resolution enhancement, the degree to which pixel intensities are directly correlated with gene expression remains uncertain (Li et al., 2024), and important limitations persist for the normal-resolution images that continue to underlie most technologies. Therefore, we chose to rely exclusively on spatial location information, thereby enhancing the usability of our method.

### 2.1 Proposed model: Square

**Hypothesis.** Tissue arrangement is generally subjected to biological patterns, such as immune infiltration or cell proliferation. Hence, cell distribution over the tissue is not stochastic, as cells belonging to the same cell type typically assemble together (Shah et al., 2016; Stoltzfus et al., 2020; Russ et al., 2021). Evidence from previous studies demonstrating spatial autocorrelation of marker genes (Ma & Zhou, 2022) further substantiates this hypothesis. Square builds on this phenomenon to increase the resolution of the spatial sample by predicting the transcriptional profile of newly generated subspots.

**Square is trained through a self-supervised learning approach**. To generate the training set, a *pooling mask* is applied by grouping $k = 9$ initial spots into super-spots by summing the transcriptional signals of the individual spots (App. Fig. 4A). This *pooling mask* is not applied on the edges where neighbors are not available. Hence, each training instance consists of the super-spot and its $m$ neighboring spots ($m = 16$ by default), creating training data with available ground truth: the location and transcriptional profile of the initial spots used in the *pooling mask*. During training, virtual nodes are used as proxy of the *higher-resolution* subspots (App. Fig. 4B). Finally, an attention mechanism is introduced into the model to predict spatially heterogeneous groups of virtual nodes based on the most relevant neighboring information for improved subspot prediction.

**Square infers unobserved subspots**. Note that at inference, data comprise the central spot to be enhanced, and its $m$ surrounding neighbors ($m = 8$ by default; App. Fig. 4C). This leads to an unavoidable distribution shift between training and inference data: the central node is much less sparse and exhibits a less blurred profile during training. Square addresses this by implementing

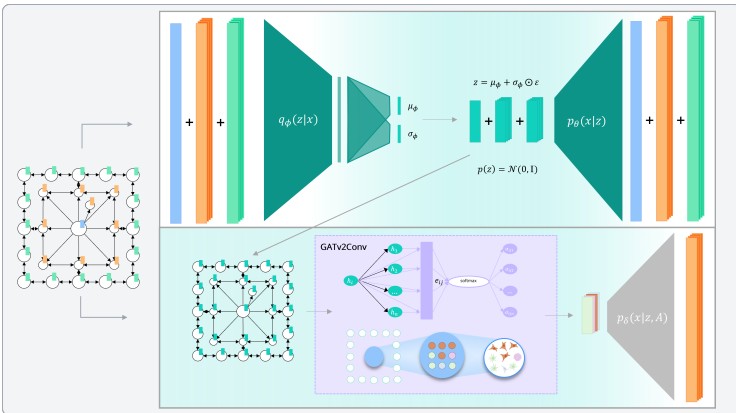

Figure 1: Square: the transcriptional profiles of the graph's nodes go through the VAE (top), and the encoded representations are used to initialize the node embeddings. Spatial dependencies learned via message passing, generating meaningful embeddings (bottom); attention is implemented using GATv2Conv (purple box); the spatially-aware embeddings are decoded back to the gene space. The depicted graph is for training, but the same steps apply at inference time.

a variational autoencoder (VAE) (Kingma & Welling, 2013), further explained below, to generate meaningful latent representations of the spots on which the sparsity signal is regressed out. We also note that the number of neighbors between training and inference differ. This is addressed by using an inductive graph-based neural network that can handle a varying number of nodes (Fig. 1).

### 2.1.1 SPATIAL CONFIGURATION USING A GRAPH-BASED REPRESENTATION

Based on the spatial topology inherent in our data, a graph $\mathcal{G}(\mathcal{V}, \mathcal{E})$ is constructed for each data instance, with the nodes representing the central spot and the surrounding $m$ neighboring spots. Following recent works on learning on graphs (Qian et al., 2024; Pham et al., 2017), we introduce $k$ virtual nodes, representing the enhanced subspots, that are connected to both the central node and the original neighbors, as well as interconnected among themselves (Fig. 4B-C). To capture the original spatial structure, edges are defined according to grid adjacency and weighted as follows:

$$A_{ij}^* = \begin{cases} d(s_i, s_j)^{-1} & \text{if } s_i \not\subset \mathcal{M} \text{ and } s_j \not\subset \mathcal{M} \text{ and } s_j \subset \mathcal{C}(s_i) \\ d(s_i, s_j)^{-1} & \text{if } s_i \subset \mathcal{M} \text{ and } s_j \subset \mathcal{M} \text{ and } s_j \subset \mathcal{C}^*(s_i) \\ t & \text{if } i = 1 \text{ and } s_j \subset \mathcal{M} \\ 1 & \text{if } s_i \not\subset \mathcal{M} \text{ and } s_j \subset \mathcal{C}^*(s_i) \\ 0 & \text{otherwise} \end{cases} \tag{1}$$

where $d(s_i, s_j)$ is the Euclidean distance between the coordinates of spots $s_i$ and $s_j$, with $s_1$ representing the original (during inference) or superspot (during training) central node. The coordinates of the virtual nodes are located between the central and neighboring nodes, specified in $\mathbf{Z} \in \mathbb{R}^{2 \times S}$, to mirror the original topology. The set $\mathcal{C}(s_i)$ contains the contiguous spots to $s_i$, $\mathcal{C}^*(s_i)$ the contiguous virtual nodes $s^*$ to $s_i$, and $\mathcal{M}$ contains the virtual spots. $t \geq 1$ is a model hyperparameter.

### 2.1.2 DISTRIBUTION SHIFT BETWEEN TRAINING AND INFERENCE DATA

The distribution shift between superspots and standard spots may hinder the generalization capabilities of the model when inferring subspots. Probabilistic frameworks have being used for different applications in omics data (Lopez et al., 2018; 2021; Bergenstråhle et al., 2022). These models are able to naturally account for the uncertainty, sparsity, and noise inherent to this type of data. Therefore, to circumvent the caveats of different resolution levels in the data, we have implemented a variational autoencoder (VAE) that learns a smooth, low-dimensional representation of the input (super)spot data while accounting for biological and technical variability. Specifically, we use a standard VAE architecture (Lopez et al., 2018) where both encoder $q_\phi(\mathbf{z}|\mathbf{x})$ and decoder $p_\theta(\mathbf{x}|\mathbf{z})$ are parametrized as neural networks with non-linear activations, and the latent space is regularized to-

wards a standard normal distribution. This architecture helps capture the dominant biological signal while smoothing over minor variations and batch effects that are not critical for the task at hand.

Thus, the nodes from the aforementioned graph $\mathcal{G}$ will have the VAE-encoded latent variables $\mathbf{z} \in \mathcal{R}^d$ as node features ($d$ is set to 50% of the number of genes). Note that the encoding process is agnostic of the graph structure, so features of a spot are independent of its neighbors'. The graph $\mathcal{G}_{\mathbf{z}_\phi}$ is populated as follows:

$$\mathbf{z}_\phi = \{\mathbf{z}^{(i)}\}_{i=1}^{1+k+m} \sim q_\phi(\mathbf{z}^{(i)}|\mathbf{x}^{(i)}), \tag{2}$$

where $k$ and $m$ are the numbers of virtual nodes and the central spot's initial neighbors, respectively.

### 2.1.3 GRAPH ATTENTION NETWORK

Square builds on the capacity of Graph Neural Networks (GNNs) to infer novel embeddings for each node by leveraging both the topology of the network as well as the features of neighboring nodes. Specifically, Square relies on a Graph Attention Network (GAT) (Brody et al., 2021), a specific type of GNN that computes an attention coefficient to infer neighbors' effect on the loss function. Using this architecture, uniform contribution of all neighbors $\mathcal{N}(i)$ of a node $i$ is avoided, learning instead a weighted average of the representations of $\mathcal{N}(i)$. This upgrade is computed via an edge scoring function $e : \mathbb{R}^d \times \mathbb{R}^d \to \mathbb{R}$ that represents the importance of each one of the neighbors of a node:

$$e(\boldsymbol{h}_i, \boldsymbol{h}_j) = \text{LeakyReLU}\left(\boldsymbol{a}^{\mathrm{T}} \cdot [\boldsymbol{W}_\delta \boldsymbol{h}_i \parallel \boldsymbol{W}_\delta \boldsymbol{h}_j]\right), \tag{3}$$

where $\boldsymbol{a} \in \mathbb{R}^{2d'}$ represents a new attention matrix, $\boldsymbol{W}_\delta \in \mathbb{R}^{d' \times d}$ stands for the standard GNN (learnable) parametric function, and $\parallel$ denotes concatenation. These edge scores (after normalization across all neighbors $j \in \mathcal{N}(i)$) are then multiplied by the matrix product of $\boldsymbol{W}_\delta$ and the neighbors' features $\boldsymbol{h}_j$. Ultimately, the update of node features after $l + 1$ layers is computed as follows:

$$\boldsymbol{h}_i^{(l+1)} = \sigma\left(\sum_{j \in \mathcal{N}(i)} \text{softmax}_j\left(e\left(\boldsymbol{h}_i^{(l)}, \boldsymbol{h}_j^{(l)}\right)\right) \cdot \boldsymbol{W}_\delta \boldsymbol{h}_j\right). \tag{4}$$

Note that equation 4 is tailored for the sum-aggregation scenario, but many others are supported. Indeed, aggregating both vectors with an element-wise maximum function has been observed to perform better in our problem. As a last step, the updated embeddings of the virtual nodes are passed through a multi-layer perceptron (MLP) to transform them to the gene space.

**Virtual nodes' feature initialization.** We use a different GNN designed to predict dataset-specific spots in between others. A prior training is performed by generating 100 different graphs with the whole dataset grid, randomly removing 20% of the spots, which are used to train the network with MSE as loss. The GNN contains 2 layers and the number of neurons at the final layer equals the number of genes. Training is conducted using SCD and a learning rate scheduler with a patience parameter of 5 epochs. This network is then used to populate the transcriptional profile of virtual nodes, which are also embedded into the latent space using the encoder $q_\phi(\mathbf{z}|\mathbf{x})$.

### 2.2 TRAINING

The VAE produces latent representations for all nodes (i.e., original spots, superspots, and virtual nodes), which are then used by the GAT to learn spatial dependencies across the graph. Given that the VAE (i.e., the encoder and decoder with parameters $\phi$ and $\theta$, respectively) is trained jointly with the GAT (with parameters $\delta$), the model needs to learn to maximize two log-likelihoods: one for the decoder distribution $p_\theta(\mathbf{x}|\mathbf{z})$, and another for the enhancement $p_\delta(\mathbf{x}|\mathbf{z}, \mathbf{A})$. Since the variational inference approach requires minimizing at the same time the distance between the encoded distribution $q_\phi(\mathbf{z}|\mathbf{x})$ and the prior $p(\mathbf{z})$, the function we maximize is:

$$\mathcal{L}_{\phi,\theta,\delta} = \mathcal{L}_{\phi,\theta}^{\text{recons}} + \mathcal{L}_{\phi,\delta}^{\text{deconv}} - \mathcal{L}_\phi^{\text{KL}}, \tag{5}$$

with

$$\mathcal{L}_{\phi,\theta}^{\text{recons}} = \mathbb{E}_{q_\phi(\mathbf{z}|\mathbf{x})}[\log p_\theta(\mathbf{x}|\mathbf{z})]; \mathcal{L}_{\phi,\delta}^{\text{deconv}} = \mathbb{E}_{q_\phi(\mathbf{z}|\mathbf{x})}[\log p_\delta(\mathbf{x}|\mathbf{z}, \mathbf{A})]; \mathcal{L}_\phi^{\text{KL}} = D_{KL}(q_\phi(\mathbf{z}|\mathbf{x})\|p(\mathbf{z})).$$

Note that $p_\delta(\mathbf{x})$ represents the probability of the observed spots' transcriptional profiles used to pseudobulk for ground truth availability. Instead, $p_\theta(\mathbf{x})$ is not restricted to a specific resolution level, but admits any genetic profile the model might process.

We use the Adam optimizer (Kingma & Ba, 2017) with a learning rate scheduler with a patience parameter of 10 epochs. Batch size has been observed to affect training performance; intuitively, it should be big enough for every batch to comprise several cell types and improve generalization.

## 3 RELATED WORK

Current methods for increasing the resolution of spatial transcriptomic data —including only those that predict genetic profiles without using any reference— rely on different solutions:

**BayesSpace** (Zhao et al., 2021) is a Bayesian statistical method designed to perform clustering analysis of spatial transcriptomic data through the integration of spatial neighborhood information with a low-dimensional representation of gene expression. BayesSpace workflow starts with a spatial clustering; these clusters then undergo a refinement process, which aims to generate a more detailed spatial map. Subsequently, the refined clustering results are translated back into the gene expression space using regression models, with principal components serving as predictors.

**CARD** (Ma & Zhou, 2022) is a spatially-informed cell-type deconvolution method that, built upon a non-negative matrix factorization (NMF) framework, models cell-type distributions by incorporating spatial dependencies using a conditional autoregressive (CAR) model. This enables both the imputation of unobserved distributions at unmeasured locations and the construction of refined maps at higher resolution. The latter allows for a resolution enhancement, allowing hence a direct comparison to Square. Although it was crafted assuming the availability of a single-cell reference, a reference-free modality has been leveraged herein for the sake of fairness.

**STAGE** (Li et al., 2024) employs an autoencoder that maps the gene expression manifold to a location manifold, which can later be used to decode previously unseen locations. Initial spots are not altered, but new spots are generated in-between. This method was envisaged for several applications (e.g., recovery of down-sampled data and 3D generation) but is herein used for reconstructing non-sequenced regions (i.e., regions between spots) to mimic Square's operation. We note that the level of refinement cannot be modified, generating always 4 subspots where one was placed originally.

## 4 EVALUATION FRAMEWORK

### 4.1 DATASETS AND DATA PREPROCESSING

**Synthetic pseudo-phenotype dataset**. This dataset is derived from 5 distinct pseudo-phenotypes that emulate the role of cell-types (App. Fig. 5A,B). We generated a fully dense $90 \times 90$ square grid, mimicking that of ST. Each spot is composed of only one pseudo-phenotype, with its gene expression profile (of dimension 100) sampled from the corresponding pseudo-phenotype's distribution (see App. A.1.1). To generate a ground truth, groups of $3 \times 3$ spots (without overlap) are further aggregated into pseudospots. While this dataset may not be realistic in practice, it serves as a baseline for method evaluation and comparison.

**Synthetic Mouse Olfactory Bulb (MOB) dataset**. We use the statistical simulator scDesign3 (Song et al., 2024), using as reference a 10X Chromium scRNA-seq (Tepe et al., 2018) and a ST spatial RNA-seq (Ståhl et al., 2016) datasets, both from a MOB sample. The dataset comprises 278 spots, each with a gene expression vector of dimension 182, and containing a mixture of up to four cell-types: granule cell, periglomerular cell, mitral and tufted cell, and olfactory sensory neuron. While we do not have access to high-resolution spots in this case, we expect the reconstructed spots to maintain the cell-type proportions of the original ones.

**Human Pancreatic Ductal Adenocarcinoma (PDAC) dataset**. We considered a ST Human PDAC dataset (Moncada et al., 2020) which contains 428 spots, each estimated to contain approximately 20-70 cells. For each spot, approximately 2,400 UMIs and 1,000 unique genes were detected. The tissue section we selected contains 20 cell-types, and mainly four regions: cancer cells and desmoplasia, nonmalignant duct epithelium, stroma, and normal acini-rich pancreatic tissue.

**Data preprocessing:** Raw transcript counts undergo three main steps: i) normalization, to remove technical biases and adjust for differences in the sequencing depth of different spots; ii) logarithmization, to stabilize the variance across the range of expression values and reduce skewness; and iii) selection of the most highly variable genes (1,000 by default), to remove the inherent noise of house-keeping and uniformly-expressed genes, as well as to reduce the number of model parameters.

## 4.2 EVALUATION METRICS

We devise several quantitative metrics that help in evaluating the effectiveness of Square in different scenarios. The metrics are reported across 5 runs to account for model's robustness.

**Ground truth:** We compute spot-wise metrics that directly compare the transcriptional profile of the generated subspots with that of the original ones. We consider the Mean Squared Error (MSE) as well as the rank-based Spearman correlation distance (SCD), the latter to measure the preservation of the relative ranking of highly expressed genes. To further evaluate whether the most highly expressed genes are retained after resolution enhancement, we compute two metrics based on the Jaccard distance (JD) between the sets of top-expressed genes in the ground truth and the reconstruction: for the genes above the mean ($JD_\mu$) and for the top 30 most expressed genes ($JD_{30}$).

We consider an additional metric to capture whether the generated spots maintain the cell-type distribution of the original spots. For each cell-type, we generate a representative by computing the mean of the corresponding single-cell transcriptional profiles. Then, the enhanced subspots are assigned the cell-type whose representative is closer in Euclidean distance to their inferred transcriptional profile. Using these predicted and true cell-type assignments, we compute the Adjusted Rand Index (ARI) to quantify the agreement between them. This metric evaluates how well the enhanced data preserves both the spatial distribution and cell-type identity compared to the ground truth.

**Cell-type proportions:** If only the cell-type proportions of the original spots are available, we consider a clustering-based assessment to determine whether the enhanced data accurately reflects the underlying cell-type distribution. We first apply $k$-means to the enhanced subspots with varying values of $k$, allowing for less and more coarse clustering. For a given $k$, we generate a *cluster-frequency* vector for every original spot, with the $i$th entry reflecting the proportion of subspots assigned to cluster $i$. Since cluster assignments may not reflect cell-types, we cannot directly compare this vector with the cell-type proportion vector. Instead, we compute the entropy of each of them, and then evaluate the Spearman correlation between them across spots. For vector $p = [p_1, p_2, \ldots, p_k]$, the entropy is computed as $H = -\sum_{i=1}^{k} p_i \log p_i$. We expect the entropies between the vectors to be correlated, as the original cell-type distribution should be reflected in the clustering assignment.

To further evaluate the enhanced subspots, we annotate each of them by first clustering the enhanced spots, and then assigning to each cluster the cell-type with more overlap between its marker genes and the cluster significative genes. A cell-type frequency vector is then constructed for each spot and compared to the ground truth cell-type proportion vector via the RMSE.

**No ground truth:** Here we rely on deconvolution method CARD (Ma & Zhou, 2022) to estimate the cell-type proportions for every original and enhanced spot. We expect the enhanced subspots to be more "pure" than the original spot, i.e., containing fewer cell-types. The intuition is that the different cell-types can be spatially distributed, and hence the inferred subspots by Square should reflect fewer cell-types based on location. In other words, we expect the entropy of the cell-type proportions of the enhanced subspots to be smaller than that of the original spots. Hence, we compute the entropies and apply a paired t-test to evaluate whether the entropy is decreased after applying Square.

## 5 RESULTS

### 5.1 ABLATION STUDY

To assess the contribution of three key components of Square to the overall performance, we conduct experiments: 1) without virtual nodes, directly updating the original nodes, and generating the reconstructed subspots by averaging the gene profiles – obtained by passing their updated embeddings through the MLP – of adjacent pairs of neighboring nodes; 2) without the VAE (not tackling the distribution shift between training and inference data), directly setting the gene expression val-

ues as the initial node features; and 3) without attention, using a standard message-passing GNN. For the assessment, we consider the *synthetic pseudo-phenotype* dataset, taking only superspots (for training) and spots (for inference) that have all neighbors available (i.e., edge spots are omitted). To better capture differences across models, we do not apply logarithmization to the data in this case.

All three ablated models underperformed Square across all metrics, except for the model without the VAE that obtained the lowest MSE (Table 1). Still, this model performs poorly on the other metrics, suggesting that the distribution shift produces a general worsening of the reconstruction if not considered thoroughly. It also exhibits the largest variance for all metrics. A notable result also emerges from visual inspection of the generated enhancements. The incorporation of attention learning introduces heterogeneity among subspot groups derived from an original spot. Without attention, the same weight matrix is propagated across all graph edges, leading to a lack of smoothness in the reconstructed expression (App. Fig. 5C). In contrast, with attention, the structural boundaries are preserved more faithfully, producing borders that closely resemble the original circular geometry rather than exhibiting a stepped or block-like pattern (App. Fig. 5D).

Table 1: **Ablation Study:** Performance comparison across model variants.

| Model | MSE $\downarrow$ | SCD $\downarrow$ | JD$_{>\mu}$ $\downarrow$ | JD$_{30}$ $\downarrow$ | ARI $\uparrow$ |
|---|---|---|---|---|---|
| w/o virtual nodes | $9.48 \pm 0.05$ | $0.441 \pm 0.011$ | $0.467 \pm 0.002$ | $0.8137 \pm 0.0005$ | $0.73 \pm 0.03$ |
| w/o VAE | $\mathbf{9.19 \pm 0.27}$ | $0.516 \pm 0.029$ | $0.488 \pm 0.015$ | $0.8161 \pm 0.0013$ | $0.64 \pm 0.10$ |
| w/o attention | $9.41 \pm 0.08$ | $0.450 \pm 0.023$ | $0.466 \pm 0.008$ | $0.8141 \pm 0.0014$ | $0.67 \pm 0.09$ |
| **Square** | $9.30 \pm 0.10$ | $\mathbf{0.435 \pm 0.013}$ | $\mathbf{0.459 \pm 0.003}$ | $\mathbf{0.8132 \pm 0.0006}$ | $\mathbf{0.79 \pm 0.03}$ |

## 5.2 SYNTHETIC PSEUDO-PHENOTYPE DATASET

We compare our model with BayesSpace, STAGE and CARD. Using the *synthetic pseudo-phenotype* data with the available ground truth, we compute the previously introduced (sub)spot-wise error and distance metrics, as well as the ARI between the predicted and true cell-types. Since STAGE generates four subspots per each initial spot but the ground truth consists of nine subspots, spot-wise metrics have been tailored by averaging the closest neighbors in the ground truth. Nevertheless, we do not compute ARI metrics since averaged profiles may not capture the spatial distribution of cells faithfully (e.g., boundaries between cell-type-specific regions would be oversmoothed).

Table 2: **Benchmarking on the *synthetic pseudo-phenotype* dataset:** Performance comparison.

| Method | MSE $\downarrow$ | SCD $\downarrow$ | JD$_{>\mu}$ $\downarrow$ | JD$_{30}$ $\downarrow$ | ARI $\uparrow$ |
|---|---|---|---|---|---|
| BayesSpace | $2.33 \pm 0.00$ | $\mathbf{0.410 \pm 0.000}$ | $0.410 \pm 0.000$ | $0.8133 \pm 0.0002$ | $0.00 \pm 0.00$ |
| STAGE | $40.18 \pm 67.57$ | $0.682 \pm 0.325$ | $0.512 \pm 0.129$ | $0.8194 \pm 0.0074$ | - |
| CARD | $9.66 \pm 0.00$ | $0.431 \pm 0.002$ | $0.450 \pm 0.001$ | $0.8132 \pm 0.0005$ | $0.00 \pm 0.00$ |
| **Square** | $\mathbf{0.32 \pm 0.08}$ | $0.440 \pm 0.012$ | $\mathbf{0.408 \pm 0.005}$ | $\mathbf{0.8131 \pm 0.0018}$ | $\mathbf{0.77 \pm 0.08}$ |

Square generates more accurate enhanced subspots than the comparison methods, yielding a higher-resolution dataset (Table 2). Specifically, it obtains a significantly lower MSE, achieves the best results in JD, albeit on par with other methods, and while it does not lead in SCD, its performance is comparable to the top-performing method. In terms of ARI, Square's inferred subspots showcase a partial agreement with the true cell-type assignments, while the other methods are no better than a random assignment. Delving more into the obtained results, we observed that CARD returns very low expression values that stand far apart from the initial data range, although they keep biological information in relative terms. The latter explains the high MSE it returns compared to BayesSpace and Square. STAGE also produces very high MSE values and lacks robustness overall, showing the highest standard deviation values for all metrics. This may be due in part to the disparity between the number of initial spots and the generated ones. BayesSpace is the method with the closest performance to Square's, but is still outperformed. The most remarkable result herein stands at the cell-type assignment comparison, which portrays the exclusive capability of Square to generate feasible subspots that accurately approximate the cell types in the ground truth, while at the same time reconstructing the spatial structures in the dataset.

## 5.3 SYNTHETIC MOUSE OLFACTORY BULB DATA

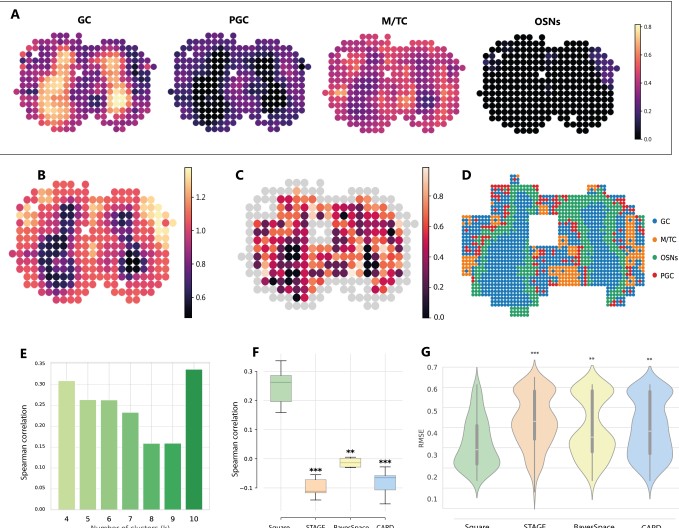

Figure 2: **A.** Cell-type abundance of the MOB synthetic dataset. **B.** Spatial entropy computed from true cell-type proportions. For Square: **C.** Cluster-frequency entropy for $k = 10$; **D.** Annotated enhanced sub-spots; **E.** Spearman correlation between the entropies of the *cell-type proportion* and *cluster-frequency* vectors for different levels of coarseness $k$. **F.** Box plots of the Spearman correlation for $k \in \{4, 10\}$. **G.** RMSE distribution of true cell-type and inferred proportions across spots.

The *synthetic MOB* dataset, which more realistically mimics a ST dataset (Fig. 2A), contains two interior areas with lower entropy spots (computed from the cell-type proportions), corresponding to the areas with mainly GC cells (Fig. 2B), and higher entropy spots located in the exterior. The entropy of the spots after enhancement is computed on the cluster-frequency vectors after applying $k$-means (see Fig. 2C for spots' entropies with $k = 10$). Only spots for which all neighbors are available are enhanced. Spearman correlations between spot entropies across coarseness levels ($k$) show a maximum of 0.33 ($k = 10$; $p$-value $< 10^{-5}$; Fig. 2E). Higher correlations are limited in part by the nature of both variables: cell-type entropy is continuous, while the cluster simplex is discrete, so the vector entropy is categorical (with even fewer possible values). This is especially notable with low-entropy spots containing a dominant cell type. Nevertheless, Square obtains significantly higher correlation values compared to STAGE, BayesSpace and CARD (Fig. 2F).

We further annotated the enhanced subspots (Fig. 2D), constructing for each spot a cell-type frequency vector that is compared to the true one. The RMSE values for Square are concentrated around 0.2, with the compared methods exhibiting bimodal distributions at higher values (Fig. 2G).

These results endorse the idea that Square is capable of untangling the mixture of different cells in the initial spots, and generating more informative phenotypes with higher-resolution spatial labels. This validation serves as a guarantee that Square is not oversmoothing or slightly jittering the initial data, but generating feasible subspots that approximate cells or smaller groups of them.

## 5.4 HUMAN PANCREATIC DUCTAL ADENOCARCINOMA

Given the lack of ground truth for this dataset, we first applied the deconvolution method CARD to the original and enhanced spots. An analysis of the entropies of the resulting cell-type proportion vectors revealed a significant decrease in the enhanced spots (Fig. 3A; p-value 0.004). The reduction in entropy suggests that Square successfully reduces the ambiguity in cell-type assignments, leading to a more accurate representation of cellular composition within the tissue.

Indeed, we further analyzed the spatial distribution of several cell-types and concluded that Square generates enhanced subspots that capture the cell-type distributions more precisely. For example, the presence of myeloid dendritic cells (mDCs) is mainly concentrated in the stroma region in both

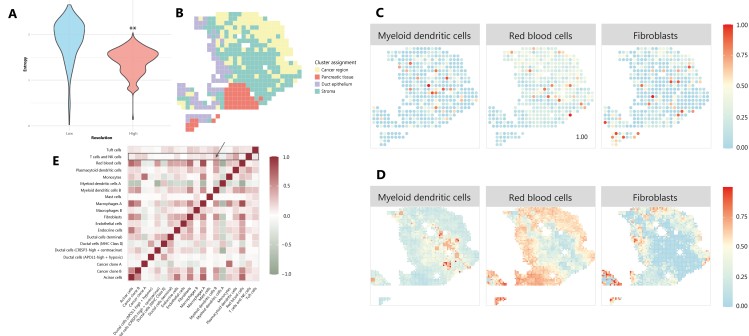

Figure 3: **A.** Entropy of the cell-type proportion vectors (computed with CARD) for the original and enhanced spots (low and high resolution, respectively). **B.** Four main regions of the spatial dataset. **C-D.** Proportion of Myeloid Dentric cells (mDCs), Red Blood cells (RBC), and Fibroblasts across the original and enhanced spots, respectively. Cell-type values do not represent absolute proportions, but normalized ones across all spots. **E.** Cell type proportion correlation computed with CARD.

the high and low-resolution data (Fig. 3B-D). However, the enhanced spots exhibit a smoother distribution and the spatial clustering of cells with identical functions. Myeloid cells are known to be enriched within the stromal regions of PDAC, and to be highly heterogeneous across tumor regions, as observed in multiplex immunofluorescence assays (Väyrynen et al., 2021). This aligns perfectly with the observations of mDCs across the sequenced space (Fig. 3D). mDCs play a complex role in the tumor microenvironment of a PDAC. In healthy conditions, they serve as an immune modulator by their antigen-presentation task; otherwise, they promote tumor growth by inducing a unique regulatory T cell program that is associated with immune tolerance and reduced survival (Barilla et al., 2019). In fact, in the enhanced data, the presence of T cells is spatially positively correlated with the presence of one of the clusters of mDCs (Fig. 3E), being the second most correlated.

Furthermore, a recent study highlights the association of PDAC with disturbances in red blood cell (RBC) aggregation (Wiewiora et al., 2023). This study confirms that PDAC is linked to excessive aggregation of RBCs, including at the pancreas cancerous regions, but not in stroma regions. We note that the distribution of RBCs after Square enhancement follows a new spatial pattern, characterized broadly by the presence of RBCs at all regions except for the stroma (Fig. 3D). Finally, a significant enrichment of fibroblasts in the ductal epithelium and the cancer region has been reported for this dataset (Moncada et al., 2020). When gazing at the CARD proportions, we see that Square is indeed deciphering this enrichment (Fig. 3B,D). The enhanced spatial distribution aligns also perfectly with the well-studied association of cancer-associated fibroblasts (CAFs) with tumor growth and immune evasion in the tumor microenvironment (Liu et al., 2019; Tao et al., 2017; Joshi et al., 2021). Furthermore, the spatial distribution of CAFs driven by this enhancement has been previously observed in different tissues (Chakiryan et al., 2021).

These analyses showcase how Square can generate biologically driven enhanced subspots using only the transcriptomic profile of the original spot and its neighbors. Applying CARD to the enhanced data yields smoother and better-localized spatial distributions. Overall, these results not only support the effectiveness of the proposed resolution enhancement technique, but also underscore its potential utility in providing finer-scale insights into the spatial organization of cell types in complex tissues, thereby enhancing our understanding of tissue architecture and function at a more detailed level.

## 6 CONCLUSION

We proposed Square, which generates high-resolution spatial data using only the spots' transcriptional profiles. Our findings demonstrate that combining graph-based models with attention mechanisms effectively disentangles mixed spatial signals and reallocates them with high confidence. Enhancing resolution not only sharpens gene expression patterns but also reveals biological insights that remain obscured under coarse spatial measurements, ultimately enabling a more precise characterization of tissue architecture and cellular diversity.

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

# A APPENDIX

## A.1 EVALUATION FRAMEWORK

### A.1.1 DATASETS AND DATA PREPROCESSING

**Synthetic pseudo-phenotype dataset.** Pseudo-phenotypes are generated from a zero-inflated negative binomial (ZINB) distribution with different parameters, i.e., the expected number of events $k$ occurring within the observed interval $\lambda$ and the zero-inflated probability $\pi$ which is fixed as the desired level of sparsity.

## A.2 ADDITIONAL FIGURES

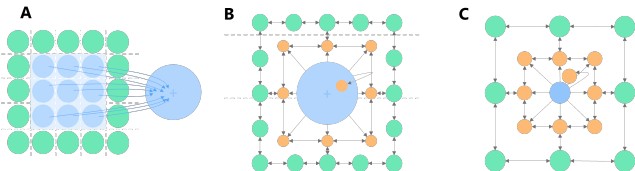

Figure 4: **A.** Training data: superspots are generated by shifting a pooling mask across the grid with overlap. **B.** Training mode: graph with a superspot (blue), its neighbors (green), and the virtual intermediate nodes (yellow). **C.** Inference mode: graph with a central spot (blue), its neighbors (green) and the virtual nodes (yellow). The virtual nodes eventually represent the enhanced subspots.

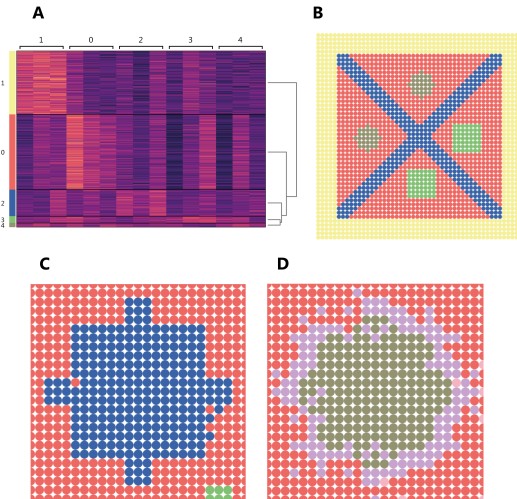

Figure 5: **A.** Heatmap of marker genes across the five generated pseudo-phenotypes. **B.** Spatial structure of pseudo-phenotypes displaying different geometries and patterns. **C.** Left circle enhancement by ablated model without attention. The colors represent cluster assignments of the enhanced subspots with $k$-means. **D.** Left circle enhancement by Square. The colors represent cluster assignments of the enhanced subspots with $k$-means.

