# OpenReview forum: "GRAPH ATTENTION NETWORK GENERATES SUPER-RESOLUTION SPATIAL TRANSCRIPTOMIC DATA"
_ICLR.cc/2026/Conference — Submitted to ICLR 2026_

### Official Review · Reviewer_rUfZ · 2025-10-30

**Soundness:** 2
**Presentation:** 2
**Contribution:** 2
**Rating:** 4
**Confidence:** 3

**Summary:**

The paper presents a computational method named Square, which is based on a Graph Attention Network (GAT) and aims to enhance the spatial resolution of spatial transcriptomics data.
 Through self-supervised learning, the method decomposes each sequencing spot into multiple subspots, thereby achieving super-resolution reconstruction of spatial transcriptomic profiles without requiring additional single-cell RNA-seq reference data.
 The core contribution of the paper lies in integrating a Variational Autoencoder (VAE) with a graph attention mechanism, which leverages information from spatially neighboring spots via message passing and attention weighting to infer subspot-level gene expression profiles.
 The authors validate the method on synthetic data, simulated mouse olfactory bulb data, and real human pancreatic ductal adenocarcinoma (PDAC) datasets, demonstrating that Square can generate subspots with more homogeneous cell-type compositions and uncover biologically meaningful spatial structures that are otherwise obscured in low-resolution data.

**Strengths:**

Combines the denoising and representation learning capabilities of the VAE with the spatial dependency modeling of the GAT, while introducing virtual nodes and a self-supervised training strategy.
 The paper is well organized and clearly presented.

**Weaknesses:**

1. The core assumption of the method is that cell distributions within tissues exhibit spatial patterns (non-random organization).
 This generally holds for most tissues, but in regions with highly mixed or extremely complex cellular organization, the model’s performance may require further validation.
2. The paper explicitly chooses not to use H&E image information, which simplifies the workflow but may also forgo valuable spatial cues.
 A more in-depth discussion of the trade-offs associated with this design choice would be beneficial.
3. Although the generated subspot data perform well on certain metrics, their impact on downstream analyses (e.g., differential expression, spatial co-expression network construction) has not been thoroughly explored.
4. The comparison with STAGE may be somewhat unequal due to the difference in the number of generated subspots; although the paper applies adjustments, this discrepancy could still affect fairness.
5. There are minor typographical errors in the manuscript.

**Questions:**

Could the authors clarify why image-informed super-resolution models such as iSTAR were not included in the comparison?
Given that iSTAR targets the same spatial transcriptomics super-resolution task but incorporates histology features, how does the proposed Square model perform relative to such image-guided approaches?

---

### Official Review · Reviewer_ncAA · 2025-10-30

**Soundness:** 2
**Presentation:** 2
**Contribution:** 2
**Rating:** 2
**Confidence:** 5

**Summary:**

This work focused on a graph attention network for generating super-resolution ST data. The authors introduced Square, the SSL method that can capture the spatial resolution of a standard sequence-based ST. It can generate up to 9 subspots per original spot and produce cellular-scale high-res data. The method includes disentangling mixed cellular signals, imputing missing transcripts, and preserving biological patterns.

**Strengths:**

- The story and motivation is clear and easy to follow.
- The method requires no reliance on external data such as H&E image which is new, as typically ST data is derived from pathology slides.
- The study show the generated ST subspots has biologically plausibility.
- This work considered the data sparsity, imputation for missing profile, which shows robust handling of real clinical data.

**Weaknesses:**

- The generation or prediction of subspots from a standard ST using Graph-based networks are not new; the authors need to discuss more on the background and related work, and compare to existing work, at least a discussion.
- The author can clarify how self-supervised learning enables real biology plausibility in the current experiment setting. The intuition for the method may lead to over-fitting due to the limited dataset size.
- The “No ground truth” evaluation needs more clarification, sounds like the results is rely on heavy assumptions and bias, does a direct validatio, such as imaging-based ST evaluation, needed?
- How does the method handle larger tissue sections or spots? How about a dynamic number of subspots?

**Questions:**

Questions and suggestions are associated with the weakness section. Thanks.

---

### Official Review · Reviewer_Vbnr · 2025-11-01

**Soundness:** 3
**Presentation:** 3
**Contribution:** 2
**Rating:** 4
**Confidence:** 4

**Summary:**

Sequencing-based spatial transcriptomics (ST) mixes multiple cells per spot, causing blur and sparsity. To address this, the authors propose Square, which boosts resolution by splitting each spot into k=9 subspots. Specifically, Square constructs a graph with both real and virtual nodes; edges follow grid adjacency. Edge weights are the inverse of spatial distance for real–real and virtual–virtual pairs, 1 for spot–adjacent virtual pairs, and a hyperparameter 𝑡 for center–virtual connections. Because training uses a 3×3 pooled superspot with 16 neighbors while inference uses the real center spot with 8 neighbors, the authors employ a VAE to mitigate the resulting distribution shift. They demonstrate that Square’s subspots yield lower cell-type entropy and sharper tissue boundaries across both real and syntehtic datasets.

**Strengths:**

The paper tackles a key limitation in this domain and proposes a detailed, well-specified graph-construction procedure to address it.

**Weaknesses:**

- Lack of novelty: The method primarily assembles existing methodologies already used in this area, with limited conceptual innovation.

- Fixed subspot count: In practice, each spot may warrant a different number of subspots, but the method fixes k and does not consider variable granularity.

- No sensitivity analysis: Hyperparameters such as t (and potentially k, neighborhood size) lack sensitivity/ablation studies.

- Attention validation: If the attention mechanism is positioned as a contribution, the paper should empirically show that attention scores align with expected structure (e.g., higher weights among same-type nodes or spatially proximate nodes).

**Questions:**

- Please provide concrete remedies for these weaknesses

- If histological images are available, what integration ideas could be combined with this method?

- Because SRT is sensitive to preprocessing, please include a sensitivity analysis on the number of highly variable genes (HVGs).

---

### Official Review · Reviewer_oSrR · 2025-11-01

**Soundness:** 2
**Presentation:** 2
**Contribution:** 2
**Rating:** 2
**Confidence:** 4

**Summary:**

This paper proposes the Square methodology, a super-resolution (or deconvolution) approach designed to resolve the inherent trade-off between gene depth and spot resolution in spatial transcriptomics data. The method operates by learning the spatial relationships between a central "anchor" spot and its neighbors within the low-resolution data. Based on this learned context, it generates virtual subspots from the single low-resolution spot to construct high-resolution data.

Square utilizes a Variational Autoencoder (VAE) framework to address the distribution shift between the data distribution during training (low-resolution) and inference (high-resolution). Furthermore, similar to prior studies, it employs a Graph Attention Network (GAT) as its backbone, allowing the model to adaptively learn from the spot's surrounding environment.

**Strengths:**

* The paper clearly explains its key motivation (addressing the trade-off between gene depth vs. spot resolution). Deconvolution and spot super-resolution are important topics in the spatial transcriptomics (SRT) domain.
* It utilizes a self-supervised learning framework instead of integrating external data like pathology images or scRNA-seq.

**Weaknesses:**

* Due to the lack of ground truth, it is not fully convincing that the generated subspots are biologically relevant. Even without ground truth, the authors could have performed a more rigorous validation of Square's plausibility by leveraging scRNA-seq or other higher-resolution information.
* Lack of Novelty: When compared to the prior work STAGE, the main differences seem to be: Handling distribution shift using VAE, generating spatial context using GAT. The novelty appears incremental.

**Questions:**

* To validate the practical utility of the generated subspots beyond the intrinsic metric of entropy reduction, we request the authors perform additional experiments benchmarked in related works like STAGE.
  * To assess model robustness, could you evaluate the recovery performance by randomly down-sampling 50% of the spots and measuring the correlation between recovered and original data?
  * To demonstrate tangible benefits for downstream analysis, could you show whether Square's high-resolution data leads to improved spatial domain identification (e.g., higher ARI) or a more robust discovery of spatially variable genes (SVGs)?

---

### Meta-Review · Area_Chair_zpnn · 2026-01-10

**Summary:**

This paper proposes a graph attention based computational method for spatial transcriptomics, aiming to enhance resolution by message passing and information sharing among neighboring spots, and to infer the cellular composition of ST spots. All four reviewers expressed overall negative assessments and raised substantial concerns, requesting clarifications and justifications across multiple aspects of the work. Unfortunately, no rebuttal or additional explanation was provided by the authors. As a result, AC finds no basis to reconsider or overturn the unanimous recommendations of the reviewers. Therefore, the final decision is Reject.

**Reviewer Concerns:**

No rebuttal was provided.

**Reviewer Scores:**

Reviewers would keep their scores unchanged.

---

### Decision · Program_Chairs · 2026-01-26

Reject